# Validation of a deep learning computer aided system for CT based lung nodule detection, classification, and growth rate estimation in a routine clinical population

John T. Murchison[1]*, Gillian Ritchie[1], David Senyszak[2], Jeroen H. Nijwening[3]*, Gerben van Veenendaal[3], Joris Wakkie[3], Edwin J. R. van Beek[1,2]

1 Department of Radiology, Royal Infirmary of Edinburgh, Edinburgh, United Kingdom, 2 Edinburgh Imaging facility QMRI, University of Edinburgh, Edinburgh, United Kingdom, 3 Aidence, Amsterdam, The Netherlands

* john.murchison@luht.scot.nhs.uk (JTM); jeroen.nijwening@aidence.com (JHN)

## Abstract

### Objective

In this study, we evaluated a commercially available computer assisted diagnosis system (CAD). The deep learning algorithm of the CAD was trained with a lung cancer screening cohort and developed for detection, classification, quantification, and growth of actionable pulmonary nodules on chest CT scans. Here, we evaluated the CAD in a retrospective cohort of a routine clinical population.

### Materials and methods

In total, a number of 337 scans of 314 different subjects with reported nodules of 3–30 mm in size were included into the evaluation. Two independent thoracic radiologists alternately reviewed scans with or without CAD assistance to detect, classify, segment, and register pulmonary nodules. A third, more experienced, radiologist served as an adjudicator. In addition, the cohort was analyzed by the CAD alone. The study cohort was divided into five different groups: 1) 178 CT studies without reported pulmonary nodules, 2) 95 studies with 1–10 pulmonary nodules, 23 studies from the same patients with 3) baseline and 4) follow-up studies, and 5) 18 CT studies with subsolid nodules. A reference standard for nodules was based on majority consensus with the third thoracic radiologist as required. Sensitivity, false positive (FP) rate and Dice inter-reader coefficient were calculated.

### Results

After analysis of 470 pulmonary nodules, the sensitivity readings for radiologists without CAD and radiologist with CAD, were 71.9% (95% CI: 66.0%, 77.0%) and 80.3% (95% CI: 75.2%, 85.0%) (p < 0.01), with average FP rate of 0.11 and 0.16 per CT scan, respectively. Accuracy and kappa of CAD for classifying solid vs sub-solid nodules was 94.2% and 0.77, respectively. Average inter-reader Dice coefficient for nodule segmentation was 0.83 (95% CI: 0.39, 0.96) and 0.86 (95% CI: 0.51, 0.95) for CAD versus readers. Mean growth

**Data Availability Statement:** Data cannot be shared publicly because of confidential patient information. Anonymous data is stored on a stand-alone server at the Edinburgh Imaging facility QMRI, University of Edinburgh, Edinburgh, UK. http://www.ed.ac.uk/edinburgh-imaging To access the data, please contact the Caldicott Guardian's Office: Caldicott Office NHS Lothian Waverley Gate 2-4 Waterloo Place Edinburgh EH1 3EG Phone +44-131-4655452 Calcicott.guardian@nhslothian.scot.nhs.uk.

**Funding:** This study was funded by NHS England via the SBRI Phase 1 grant for "Early Detection and Diagnosis of Cancer" which was granted to Aidence (Amsterdam, the Netherlands). Aidence provided support with this grant in the form of salaries for authors [JTM, GR, EJRVB], but did not have any additional role in the study design, data collection and most of the analysis, decision to publish, or preparation of the manuscript. The specific roles of these authors are articulated in the 'author contributions' section. EJRvB is a member of the Medical Advisory Board of Aidence and received support in the form of salary for the work performed in this study. JTM has no affiliation with Aidence and received support in the form of salary for the work performed in this study. GR has no affiliation with Aidence and received support in the form of salary for the work performed in this study. DS has no affiliation with Aidence and received support in the form of salary for the work performed in this study. JHN is a full time, paid employee of Aidence at the time of submission of this manuscript. GvV is a full time, paid employee of Aidence at the time of submission of this manuscript.

**Competing interests:** JTM, GR, and DS declare no competing interests. EJRvB declares ownership of QCTIS Ltd and serves on the medical advisory boards of Aidence BV and Imbio LLC. EJRvB received a restricted research grant from Siemens Healthineers and speaker fees from AstraZeneca and Roche Diagnostics. EJRvB's non-financial competing interest is serving as an expert witness on medicolegal advice on imaging based cases. JHN and GvV are full time, paid employees of Aidence and have aided in part of the raw data analysis (GvV) or re-submitting this manuscript to PLOS ONE (JHN). These interests do not alter our adherence to PLOS ONE policies on sharing data and materials.

**Abbreviations:** 3D, 3 dimensional; CAD, Computer Assisted Detection; CADe, Computer Assisted Detection Device; CADx, Computer Assisted Diagnostic Device; CE, Conformité Européenne; CI, Confidence Interval; CT, Computed Tomography;

percentage discrepancy of readers and CAD alone was 1.30 (95% CI: 1.02, 2.21) and 1.35 (95% CI: 1.01, 4.99), respectively.

## Conclusion

The applied CAD significantly increased radiologist's detection of actionable nodules yet also minimally increasing the false positive rate. The CAD can automatically classify and quantify nodules and calculate nodule growth rate in a cohort of a routine clinical population. Results suggest this Deep Learning software has the potential to assist chest radiologists in the tasks of pulmonary nodule detection and management within their routine clinical practice.

## Introduction

Lung nodule detection and management is one of the most frequent challenges in chest computed tomography (CT), not just in the context of lung cancer screening, but also in the staging of other malignancies in routine clinical practice. Lung cancer remains the third most prevalent cancer worldwide, is both rising in incidence [1], and maintains high mortality rates with around 1.8 million global deaths annually. Several recent studies demonstrated the benefits of lung cancer screening on early detection and improved outcomes [2–4]. The advent of lung cancer screening results in the need to detect smaller nodules, and therefore, the importance of fast and accurate detection is even more pronounced [5].

Lung cancer is ideally diagnosed by histopathological confirmation. However, the diagnostic process usually begins with chest CT where pulmonary nodules are identified incidentally. Pulmonary nodules are very common and mostly benign, however they should be considered as early stage cancers. The biggest challenges for pulmonary nodule detection on CT are acceptable sensitivity levels and reading times. Many failures in lung cancer diagnoses are due to detection errors rather than interpretation [6, 7]. Several studies showed that the performance of (sub-specialist) radiologists for detecting pulmonary nodules is suboptimal with reported sensitivities around 80% [8, 9].

Pulmonary nodule guidelines recommend different cut-off levels for nodule size and/or volume and volume doubling time as metrics to assess nodule size and growth [10–15]. There is increasing consensus that semi-automated volume assessment gives the most robust assessment for lung nodule growth during follow up [5, 14, 15]. Another important parameter to consider is pulmonary nodule composition (solid vs sub-solid), as sub-solid nodules are more likely to be malignant [16].

The above-mentioned challenges lead to many hospitals currently unable to assess nodules in a timely and accurate manner. Software aided detection and classification of lung nodules should improve the radiologist's diagnostic arsenal and throughput time and additionally could facilitate the roll-out of CT lung cancer screening [17]. Therefore, there has been an increasing focus on developing deep learning based computer assisted detection systems to facilitate more rapid reporting [18–28]. A few of these systems have reached availability for use in clinical practice. The study described here was performed to validate one such system, which was originally trained on a lung cancer screening cohort, in a retrospective clinical population cohort of Scottish patients undergoing routine chest CT investigations.

CTDIvol, Volume CT Dose Index; DICOM, Digital Imaging and Communications in Medicine; FP, False Positive; FN, False negative; FROC, Free Response Receiver Operating Characteristic; GE, General Electric; MIP, Maximum Intensity Projection; MPR, Multiplanar reconstruction; kVp, Kilovoltage peak; mAs, Milliamp seconds; mGy, Milligray; NLST, National Lung Screening Trial; TP, True positive; VDT, Volume Doubling Time.

# Materials and methods

## Subject selection

CT studies from a routine clinical population, in a single academic hospital, between January 2008 and December 2009 (9 years before start of this retrospective study), were searched for the following inclusion criteria: age 50–74 years, current smokers, a smoking history and/or radiological evidence of pulmonary emphysema. CT studies excluded from the analysis had slice thickness >3mm, or the presence of diffuse pulmonary disease in the radiology report, and/or the CT images, with widespread abnormalities such as interstitial lung disease.

In total, 337 fully anonymized chest CT examinations from 314 subjects (173 women, 164 men) with reported nodule size of ≥3mm and ≤30mm were included and transferred onto a stand-alone server. A waiver of informed consent was obtained from the South East Scotland Research Ethics Service.

From these CT scans, five groups were created. Group 1: 178 CT scans, initially reported as being free from pulmonary nodules. Group 2: 95 CT scans, reported to have between 1 and 10 pulmonary nodules. Group 3: 23 CT scans from patients undergoing follow-up of a pulmonary nodule. Group 4, consisted of the 23 follow-up scans of group 3. Finally, group 5 consisted of 18 scans to enrich the study group with part-solid and/or ground-glass nodule(s).

## CT protocol

A Toshiba Aquilion was used for most (330) studies; intravenous contrast was used in 22 CT scans. The mean tube peak potential energies used was 120 kVp, (range: 120–140 kVp), the average tube current was 243 mAs (range: 80–491 mAs) and the average CTDIvol was 14.0 mGy (range: 2.9–29.7). Data was reconstructed at a mean slice thickness of 1.0 mm (range 1.0–2.5mm). All CT scans were reconstructed using filtered back-projection, as these studies predated the routine application of novel reconstruction methods, such as iterative reconstruction. Other CT scanners used were: Toshiba Aquilion-CX: 2 scans, Toshiba Aquilion ONE: 1 scan, GE Medical Systems LightSpeed 16: 2 scans, GE Medical Systems LightSpeed: 2 scans.

## Nodule definition

The Fleischner Society's definition for pulmonary nodules was broadly used during this study [12]. The size range was 3–30 mm with "actionable nodules" regarded as having a largest axial diameter between ≥5mm (or a volume of ≥80mm³) and ≤30mm as recommended by the British Thoracic Society guidelines [10].

## CAD software

Veye Chest version 2.0 (now known as Veye Lung Nodules, developed by Aidence B.V., Amsterdam, the Netherlands), which is CE marked and certified as a Class IIb medical device, was evaluated in this study, see (S1 Fig). The software is primarily based on Deep Learning technology, which was trained on 45k+ chest CT-scans (slice thickness ≤3mm without contrast fluid) and 40k+ annotations by radiologists. The software runs automatically and comprises of CADe and CADx functionality and growth rate calculation. The software has a detection threshold based on nodule likelihood values (range 0.0 to 1.0). For this study the threshold was set to 0.1 which means that the threshold is set to a high sensitivity and consequently a relatively high false positive rate.

## Image annotation

A panel consisting of three thoracic radiologists ($\geq$ 9 years' experience; JTM, GR and EJRvB, expert readers 1, 2 and 3, respectively) received training on the annotation tasks and annotation tool with written instructions available throughout. The study was performed at the University of Edinburgh between February–May 2018.

Two datasets were created from the 337 CT scans: one set with CAD results and one set without CAD results. Reader 1 reviewed all the CT scans, but half of the CT scans with the CAD results (CAD aided) and the other half without CAD results (CAD unaided). For reader 2 this was vice versa. Hence, each CT scan was reviewed twice, once by one reader with the CAD results (CAD aided) and once by the other reader without the use of CAD (CAD unaided). Readers had to identify all lesions they considered to be a pulmonary nodule without clear benign morphological characteristics (calcification, typical perifissural lymph node). Any nodules requiring follow-up according to lung cancer screening criteria were classified as "actionable nodules" [10]. The Reader would mark an actionable pulmonary nodule manually on unaided scans or classify a CAD prompt on an aided scan as either true positive (TP) or false positive (FP). Any actionable nodules identified on aided scans, which had not been detected by CAD were also recorded. Readers registered all actionable nodules present on CT scans from groups 3 and 4. Finally, the readers classified all FP CAD prompts into four different groups: micro-nodules (largest axial diameter <3mm), masses (largest axial diameter >30mm), benign nodules (benign calcification pattern or clear benign perifissural appearance) and non-nodules (1088 non-nodules in total. More specific: atelectasis: 283; scar tissue: 229; fibrosis: 157; vessels: 126; non-lung: 81; other: 81; pleural: 80; fissure: 25; pleural plaque: 14; consolidations: 12 pleural plaque).

After completing all the readings on the workstations the readers reviewed their own previously identified nodules on a tablet (iPad Pro). The reader was asked to determine the composition (solid or sub-solid) and segment each nodule on every slice. The results from readers 1 and 2 were evaluated for the presence of any discrepancies. Discrepancies were defined as a difference between the results in terms of: location (3D Dice coefficient of 0); composition; segmentation (3D Dice coefficient < -1 standard deviation of the mean) and nodule registration. The Dice coefficient is a spatial overlap index and a reproducibility validation metric with a range of 0.0 (no overlap) to 1.0 (perfect overlap) [29].

Reader 3 subsequently adjudicated all discrepancies without the results of CAD using the same materials used in the blinded phase. Reader 3 created an independent reading for each nodule that had a discrepancy for at least one characteristic.

## Reference standard

The reference standard for actionable nodules consisted of lesions from groups 1 and 2 which were marked as a pulmonary nodule by the majority of the panel and met the size criteria of having a largest axial diameter between $\geq$5mm (or a volume of $\geq$80mm$^3$) and $\leq$30mm. The majority consisted of consensus between reader 1 and 2 or, in the case of no consensus, the adjudication of reader 3. The location of an actionable nodule was defined by averaging the center of mass of all reader's segmentations. Subsequently, the radius and volume were derived from these segmentations. The reference standard for composition was determined by majority consensus of lesions from groups 1–3 and 5. Finally, growth rate was determined as the relative volume difference between nodules visible on a study from group 3 and on its follow-up study from group 4.

## Data analysis

Findings from a reader or from CAD were scored as either TP, if the center of the detection was within the volume of actionable nodules in the reference standard, or otherwise as FP.

Findings from a reader or CAD in the center of the detection that was within the volume of a micro-nodule or a mass or a nodule detected by only a single reader were neither scored TP or FP. The absence of a prompt from CAD in the center of an actionable nodule in the reference standard was considered FN. Sensitivity for detecting actionable nodules and the average number of FP detections per CT scan for AIDED readings, UNAIDED readings and CAD alone was calculated using the reference standard for actionable nodules.

The sensitivity, specificity, positive predictive value and negative predictive value, accuracy and kappa score for determining the composition (solid or sub-solid) by CAD alone was calculated using the reference standard for composition.

The segmentation accuracy of readers was calculated as the Dice coefficient between each reader's segmentation and averaged (inter-reader dice coefficient). The segmentation accuracy of CAD alone was calculated as the Dice coefficient between each CAD segmentation and each individual reader segmentation and averaged. In addition, the inter-reader mean diametric and volumetric discrepancy was calculated using the largest axial diameter and volume from each segmentation of each reader's segmentation and compared to those from the other readers, this was also calculated for CAD alone compared to the other readers.

For sequential scans (groups 3 and 4), nodule registration from CAD was scored as either TP, if the detected registration was included in the nodule registration reference standard, or otherwise as FP. The mean discrepancy between growth percentages determined by readers and CAD alone was calculated.

## Statistical analysis

One-tailed Welch's t-test was used to accept the hypothesis that the mean sensitivity of AIDED is higher than the mean sensitivity of the UNAIDED readings ($p < 0.05$), with the use of bootstrapping over scans with 2000 iterations. One-tailed Welch's t-test was used to accept the hypothesis that the mean CAD Dice score is higher than the mean inter-reader Dice score ($p < 0.05$).

## Results

Groups 1 and 2 consisted of 273 CT scans with 269 actionable nodules see Table 1. Remarkably, nodules were identified in group 1, highlighting the importance of concurrent reading. The radiologists with CAD readings showed a sensitivity of 93.5% and average FP rate of 3.0. The sensitivity for detecting actionable nodules of radiologists without CAD on scans from groups 1 and 2 was: 71.9% (95% CI: 66.0%, 77.0%) and 80.3% (95% CI: 75.2%, 85.0%) ($p < 0.01$), respectively. The average FP rate of radiologists alone and radiologists with CAD readers was: 0.11 and 0.16, respectively. The maximum obtainable sensitivity of CAD alone was 95.9% at an average FP rate of 10.9. The sensitivity of CAD alone was equivalent to

**Table 1. Distribution of study subjects and nodule size by group.**

| Group | Number of subjects | Number of CT scans | Number of nodules with largest axial diameter $\geq 3$ and $<5$mm | Number of nodules with largest axial diameter or mean volume $\geq 5$mm / $\geq 80$mm$^3$ and $<30$mm |
|---|---|---|---|---|
| 1 | 178 | 178 | 19 | 71 |
| 2 | 95 | 95 | 34 | 198 |
| 3 | 23 | 23 | 0 | 68 |
| 4 | | 23 | 6 | 36 |
| 5 | 18 | 18 | 2 | 36 |
| **TOTAL** | **314** | **337** | **61** | **409** |

radiologists without and radiologists with CAD readings at an average FP rate of 0.62 and 0.88, respectively (Fig 1). Details regarding the number of CT scans and nodules per group are described in Table 1.

The composition of nodules within groups 1, 2, 3, and 5 totaled 325 solid nodules and 57 sub-solid nodules. The sensitivity, specificity, positive predictive value and negative predictive value of CAD for determining the composition of solid nodules in groups 1, 2, 3, and 5 was 98.8%, 68.4%, 90.7% and 94.7%, and was 68.4%, 98.8%, 94.7% and 90.7% for sub-solid nodules, respectively. The accuracy and kappa of CAD for determining the composition (solid or sub-solid) of a pulmonary nodule was 94.2% and 0.77.

The CAD software successfully segmented 95% of pulmonary nodules from groups 1–3 and 5. The average inter-reader Dice coefficient was 0.83 (95% CI: 0.39, 0.96) versus 0.86 (95% CI: 0.51, 0.95) for CAD alone (p<0.01). The mean largest axial diameter of all nodules was 7.68 ± 3.50 mm (range: 3.42–28.45 mm) and the mean volume was 198 ± 333 mm3 (range: 21–2797 mm$^3$. The inter-reader geometric mean diameter discrepancy was 1.15 (95% CI: 1.00, 1.58) versus 1.17 (95% CI: 1.01, 1.69) for CAD alone. The inter-reader geometric mean volumetric discrepancy was 1.39 (95% CI: 1.01, 3.19) versus 1.38 (95% CI: 1.01, 3.38) for CAD alone.

The total number of nodules in group 3 and 4 was 68 and 42, respectively. The total number of nodule-pairs in groups 3 and 4 was 23 and all nodules were successfully identified by CAD.

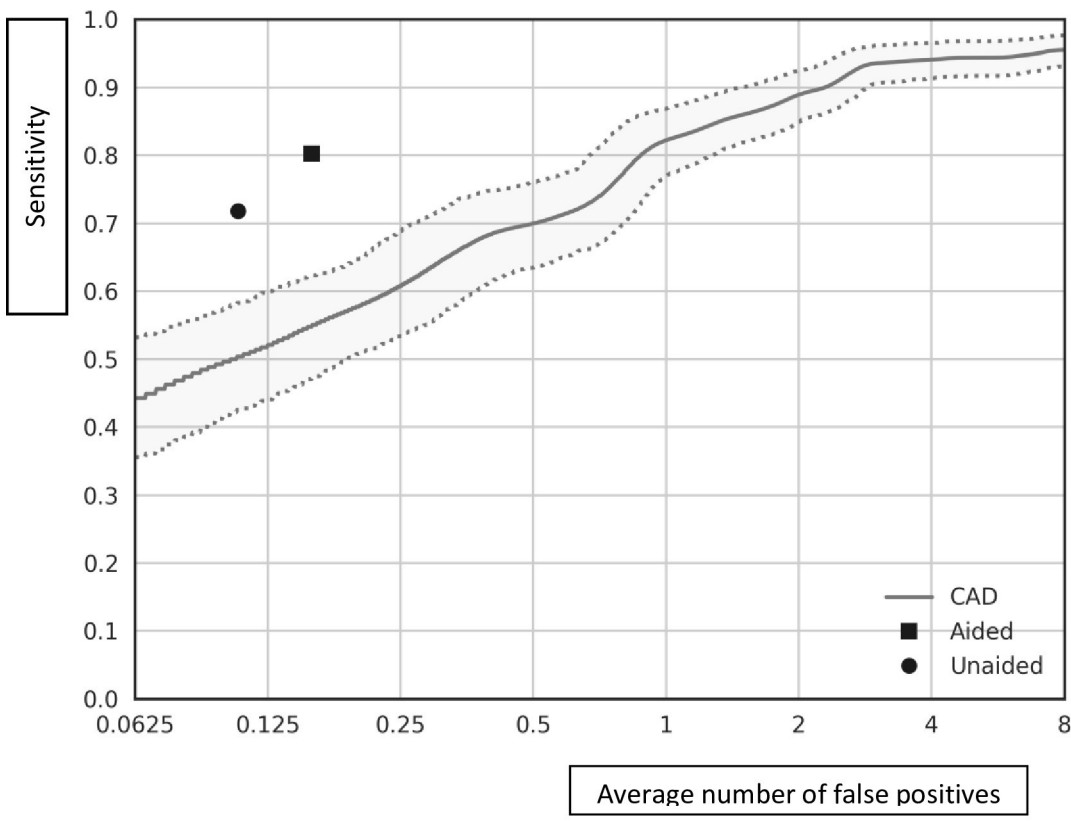

**Fig 1. Free-response ROC (FROC) curve.** This curve shows the standalone performance of CAD for detecting actionable nodules based on scans from groups 1 and 2. The vertical axis represent the sensitivity and the horizontal axis the average number of false positives per scan. The dashed lines show the upper and lower boundary of 95% confidence interval, bootstrapping of scans with 2000 samples. The circle represents the UNAIDED performance (sensitivity: 71.9% average FP rate 0.11 per scan) and square the AIDED performance (sensitivity: 80.3% average FP rate 0.16 per scan) for detecting actionable nodules.

The mean growth percentage discrepancy of readers and CAD alone was 1.30 (95% CI: 1.02, 2.21) and 1.35 (95% CI: 1.01, 4.99), respectively, which was not statistically significant.

## Discussion

The study described here shows improved sensitivity of experienced thoracic radiologists using aided detection from 71.9% to 90.3% with a minor increase in FP rate. The maximum stand-alone CAD sensitivity was 95.9% at an average FP rate of 10.9, which would be unworkable in clinical practice. A more acceptable average FP rate would be between 1 and 2 with corresponding sensitivity range (82.3% - 89.0%), outperforming thoracic radiologists with and without using CAD. The standalone performance of the CAD, when set to the threshold of 0.1 applied in this study, correlates to an average sensitivity of 95% and an average number of 7 false positives per study based on this dataset.

Computer assisted detection and diagnosis software, including convolutional neural networks and machine learning approaches have shown promising results in aiding radiologists to identify incidental pulmonary nodules. A study using the LIDC database as a comparison tested 108 CT scans and demonstrated high sensitivity and specificity [18, 20]. However, there are also conflicting results. A more recent study [21] demonstrated moderately high sensitivity of 84% and a corresponding positive predictive value of 67% when tested in 100 patients with 106 biopsied lung nodules at a slice thickness of 3 mm. Another commercial system was clearly suboptimal when tested on 50 pure ground glass and 50 part solid nodules [22]. The most comprehensive deep learning system to date used 11,625 chest CT scans for model training and validation and subsequently used 1,129 chest CT studies for testing of the model with a sensitivity between 74%-86% at FP rates of 1–8, respectively [23].

This is the first study using this CAD software to look at a routine cohort of smokers who underwent chest CT for non-screening purposes. The software tested here was initially validated on a lung cancer screening population [2, 17] and the results of our study are of similar sensitivity and accuracy to that initial cohort, (87% at 1 FP/scan) [2] confirming broader use is feasible.

In this study, AIDED readings outperformed UNAIDED readings, yielding a sensitivity of 93.5% at an average FP rate of 3.0. However, 36 CAD detected nodules confirmed by the majority of the panel were scored as FP by one reader. A possible explanation could be that due to the high number the readers develop a tendency to call CAD prompts FP. Another explanation could be a structural difference in pulmonary nodule definition between the readers. Even allowing for this, the number of TP nodules detected by CAD was higher than without CAD.

For determining the composition (solid or sub-solid) of a pulmonary nodule, the CAD software yielded a high accuracy of 94.2% and a kappa score of 0.77. The segmentation accuracy of CAD was similar to that of thoracic radiologists, CAD dice 0.86 and inter-reader dice 0.83 (p <0.01).

In addition, the CAD software yielded a perfect score for a limited number of nodule pairs and analyzing its volumes; sensitivity 100.0% without FP pairs, but further validation will be required. The mean growth percentage discrepancy of readers was 1.30 compared to 1.35 for CAD alone. However, due to a single incorrect segmentation of the CAD software, the upper end of its confidence interval (95% CI,1.01–4.99) is twice as high compared to that of readers (95% CI,1.02–2.21), illustrating that visual verification is still required. Nevertheless, this compares favorably with results from a software comparison sub-study of the NELSON study in 50 subjects [25]. Similarly, a study of 134 participants in the NLST also demonstrated a decrease in variability of detection and volumetry with the use of software [26].

This study has several limitations. First, the data was obtained from a single site and the vast majority of CT scans were acquired by a single CT scanner vendor. A recent study demonstrated decreased diagnostic performance of machine learning-based radiomics models in 26 patients with subsolid adenocarcinoma nodules when iterative reconstruction was applied [27]. Therefore, care must be taken to validate any software tool on actual datasets. Although differences between scanner manufacturers and CT imaging protocols may alter the interpretation of lung parenchymal features, it is unlikely to significantly affect the presence or absence of actionable pulmonary nodules. Indeed, all vendors have taken part in various CT lung screening trials and have shown similar results. Second, the readings were performed under artificial conditions and therefore the performance of the CAD software and the radiologists may be different in a real-world setting. This is considered of potential importance, as artificial conditions and use in selected datasets tend to lead to excellent results of lung CT CAD systems [28, 29]. Further prospective clinical validation is therefore required, and this also highlights the need for seamless workflow integration of this software for it to become standard practice. Lastly, the sensitivity of the readers without and with CAD versus CAD alone was calculated using the reference standard established by the same readers and CAD. The only addition to this was the third reader, who effectively assured consensus on the final classification and morphologic features of lung nodules. One could consider performing the same test in multiple readers, but this would be time consuming and unlikely lead to significantly different results. Recently, the software described here was independently evaluated in a large teaching hospital [30]. This study found a sensitivity of 88% and the mean FP rate was 1.04 FPs per scan.

In conclusion, the use of the CAD significantly increased radiologist's detection of actionable nodules yet also increasing the false positive rate. The Deep Learning model for nodule detection was trained on data from a lung cancer screening cohort. In addition, this study appears to show that it is also effective in a general, "real life" clinical setting where it improves the sensitivity of detection of actionable nodules by thoracic radiologists. This CAD system is able to automatically classify, quantify, and calculate the growth rate of pulmonary nodules. These results suggest that Deep Learning software has the potential to assist radiologists in the tasks of pulmonary nodule detection and management on routine chest CT.

## Supporting information

**S1 Fig. Screenshot of Veye Chest.**
(TIF)

## Author Contributions

**Conceptualization:** John T. Murchison, Edwin J. R. van Beek.

**Data curation:** Gerben van Veenendaal.

**Formal analysis:** Gillian Ritchie, David Senyszak.

**Investigation:** John T. Murchison, Gillian Ritchie, David Senyszak, Edwin J. R. van Beek.

**Methodology:** John T. Murchison, Gillian Ritchie, David Senyszak, Joris Wakkie, Edwin J. R. van Beek.

**Supervision:** John T. Murchison, Edwin J. R. van Beek.

**Writing – review & editing:** John T. Murchison, Gillian Ritchie, David Senyszak, Jeroen H. Nijwening, Joris Wakkie, Edwin J. R. van Beek.

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
