## [Decision Letter · Decision Letter 0]

12 May 2021

PONE-D-21-07234

Validation of a deep learning computer aided system for CT based lung nodule detection, classification and quantification and growth rate estimation in a routine clinical population

PLOS ONE

Dear Dr. Nijwening,

Thank you for submitting your manuscript to PLOS ONE. After careful consideration, we feel that it has merit but does not fully meet PLOS ONE’s publication criteria as it currently stands. Therefore, we invite you to submit a revised version of the manuscript that addresses the points raised during the review process.

We look forward to receiving your revised manuscript.

Kind regards,

Chang Min Park, MD, Ph.D

Academic Editor

PLOS ONE

Journal Requirements:

"Prof. Van Beek is a member of the Advisory Board of Aidence.

Prof. Murchison, Dr. Ritchie and Mr. Senyszak declare no interest. "

We note that one or more of the authors have an affiliation to the commercial funders of this research study : Aidence.

2.1. Please provide an amended Funding Statement declaring this commercial affiliation, as well as a statement regarding the Role of Funders in your study. If the funding organization did not play a role in the study design, data collection and analysis, decision to publish, or preparation of the manuscript and only provided financial support in the form of authors' salaries and/or research materials, please review your statements relating to the author contributions, and ensure you have specifically and accurately indicated the role(s) that these authors had in your study. You can update author roles in the Author Contributions section of the online submission form.

2.2. Please also provide an updated Competing Interests Statement declaring this commercial affiliation along with any other relevant declarations relating to employment, consultancy, patents, products in development, or marketed products, etc.  

4. Please include your tables as part of your main manuscript and remove the individual files. Please note that supplementary tables should be uploaded as separate "supporting information" files.

Reviewers' comments:

Reviewer's Responses to Questions

**Comments to the Author**

1. Is the manuscript technically sound, and do the data support the conclusions?

Reviewer #1: Yes

Reviewer #2: Partly

2. Has the statistical analysis been performed appropriately and rigorously? 

Reviewer #1: I Don't Know

Reviewer #2: No

3. Have the authors made all data underlying the findings in their manuscript fully available?

Reviewer #1: Yes

Reviewer #2: No

4. Is the manuscript presented in an intelligible fashion and written in standard English?

Reviewer #1: No

Reviewer #2: Yes

5. Review Comments to the Author

Reviewer #1: 1. Abstract: The important points are well organized in the abstract.

2. Introduction: It reflected well the reality and difficulties of detecting pulmonary nodules on CT scans. In addtion, the authoes well explanied the importance of CAD regarding this point.

3. Subject selection - exclusion criteria: the authors excluded diffuse pulmonary disease such as ILD. What threshold they used for this exclusion? If patients had very subtle ILD (or ILA), were they excluded? And how could the authors know the existence of ILD? They reviewed all the CT exams? Please clarify.

5. Subject selection- group categorization seems cumbersome for future readers. and the authors decribed that Group 1 consisted of 178 CT scans reported as being free from pulmonary nodules. But the result (Table 1) is different. It was read that there was no pulmonary nodule, but were these cases actually had pulmonary nodules?

6. Only one CT machine was used in this study. If so, it seems very unified protocols and well organized study protocol. If not, please state more clear about types of CT machines

7. Nodule definition - please consider to delete "“pulmonary nodule” was not firmly defined since the notion of nodule may not represent a single entity capable of verbal definition." It seems unnecessory.

8. Nodule definition - the authors defined "actionable nodules" as a largest axial diameter between ≥5mm (or a volume of ≥80mm3) and ≤30mm. I wonder whether they have some references for this definition or they arbitraly decided it.

9. CAD software - For readers who are not friendly with this commercially avilable CAD system, please add detailed information about the CAD. For example, 1) why the threshold was decided as 0.1? what dose it mean? 2) which CT examinations were used for its development in terms of slice thickness or use of contrast media?

10. Image annotation - "three different groups: micro-nodules, masses, benign nodules and non-nodules." Not three different groups. Please revise it.

11. Reference standard and Data analysis -The description is rather complex and difficult to understand. I hope it is organized so that it is easy to understand.

12. Reference standard and Data analysis -The most curious thing is that the authors created a reference standard as a result of the panels consisting of readers 1, 2, and 3. Nevertheless, it seems that the performance evaluation of readers was done as this reference standard.

13. Results: The content of the result may seem appropriate, but revision should be made according to the above items being modified.

Reviewer #2: This paper describes a retrospective validation study of a deep learning computer-aided diagnosis system for lung nodule detection, classification, quantification and growth rate estimation in a routine clinical population. For this study, a retrospective dataset from one academic center in Scotland is collected with scans made in 2008 and 2009. In total, 337 scans from 314 different subjects are collected. If I understand correctly, the original radiology reports of these scans are consulted and based on these reports, the scans are divided over 5 different groups.

A panel of 3 radiologists is used to annotate the dataset where the readers alternated between an aided and unaided reading. So, each reader read half of the scans with CAD, and half of the scans without CAD. A third experienced reader finally resolved discrepancies between the two readers.

The paper shows the performance of CAD was better than the radiologists on this dataset, and showed good results for segmentation, growth rate estimation and nodule type classification.

The authors conclude that the CAD system significantly increased the detection performance of radiologists for actionable nodules while only minimally increasing the false positive rate.

I have several major comments of criticism:

- The CAD system that is under investigation here is used for setting the reference standard. In addition, the readers for which the aided and unaided performance is reported, are used to make the reference standard. This affects the results and potentially positively biases the CAD results. The authors have also mentioned this in the Discussion so it is recognized as a limitation already. I think the conclusions of this study should therefore be less strong. It would be best if two other readers would also split all cases and read all of them without CAD support. Then, a truly independent read would be available. Ideally, this reader would read with and without CAD, but then, all cases need to be read twice, so is more effort.

- Related to the first point: A proper experiment setup for an observer study to compare aided vs unaided reads of data is multi-reader multi-case (MRMC) analysis, which also allows for better statistical comparisons than the t-tests that are performed now. If it is still possible to ask additional readers, that would make the study stronger. In addition, please consult with a statistician to see whether a form of MRMC analysis is possible on this data.

- The CAD is used at a setting with high sensitivity and relatively high false positive rates. If this is the setting that the radiologists used while reading the cases, then the paper should also report the standalone performance of the CAD system at this operating point. It is not clear to me what the performance is at this operating point. Several operating points are reported by the authors, but it is not clear to me which one correspond to the setting used by the readers (threshold of 0.1). Please add that as a dot on the FROC curve in Figure 1. Please also discuss what effect this setting may have on the study results in the Discussion part.

- Is the current CAD system approved/cleared for clinical use as a second reader or concurrent reader? This is not clear to me.

- Some important related literature in this area is not covered, for example: https://www.ajronline.org/doi/pdf/10.2214/AJR.17.18718

- A table which gives a breakdown of the CAD marks and the TP and FP categories would be very useful. The authors wrote "Finally, the readers classified all FP CAD prompts into three different groups: micro-nodules (largest axial diameter <3mm), masses (largest axial diameter >30mm), benign nodules (benign calcification pattern or clear benign perifissural appearance) and non-nodules (pleural plaque, scar tissue, atelectasis, fibrosis, fissure thickening, pleural fluid, pleural thickening, intrapulmonary vessels, consolidations, outside of lung tissue, or other (free format))." So, for every CAD mark, this rating is available so would be great to see a table with this information.

Detailed comments:

- Please be more clear about the selection of the CT cases. If I understand correctly, the original radiologist reports were manually checked to see whether nodules were reported or not, and that gave groups 1 and 2. But then the abstract should not state the exact nodule numbers yet, because I suppose this is the result after the annotation process, so should be in the Results section of the abstract.

- The statement on Data Availability: I understand that the scans cannot be shared, but it would be good if the result files with the readings can be shared. So, a csv file with for each cases the recorded nodules per reader, and whether they were in the end part of the reference standard, etc.

- How many CAD marks in total were there at the threshold of 0.1?

- How many CAD marks were there on micronodules or masses or marks only annotated by one reader and thus ignored in the FROC analysis? Please report.

6. PLOS authors have the option to publish the peer review history of their article (what does this mean?). If published, this will include your full peer review and any attached files.

Reviewer #1: No

Reviewer #2: No

---

## [Author Response · Author response to Decision Letter 0]

16 Sep 2021

Response to reviewers

Dear PLOS editorial team, reviewer #1 and reviewer #2, 

Thank you for your comments, critical questions and suggestions. We have updated the manuscript accordingly and have addressed your comments in the response letter below. 

Our apologies for our late reply. The reason we give for this is that the software analyzed in this study is developed by a start-up company with limited resources. Most of our efforts in the past months have gone to additional studies that are required to obtain 510(k) approval from the FDA. 

Our CAD has been further developed since the study described in this paper and has increased in performance, both in sensitivity and specificity. Yet, we believe it is still valuable for the scientific literature to publish our initial clinical validation of the software. We believe that transparency is one of the most fundamental values in science. 

Kind Regards,

Jeroen Nijwening

Response to Academic editors

Dear academic editors, we have adapted the style of the manuscript as requested. Furthermore, we have included a new cover letter which contains the updated information regarding the funding statement and competing interest statement. 

Response to Reviewer #1

1. The important points are well organized in the abstract.

Thank you for your compliment regarding the abstract. We have revised the abstract based on the reviewers’ suggestions and expect that it still holds the core message of the manuscript.

2. Introduction: It reflected well the reality and difficulties of detecting pulmonary nodules on CT scans. In addition, the authors well explained the importance of CAD regarding this point.

Thank you for highlighting the importance of CAD within a radiologist’s practice. We believe that CAD will improve the quality of reports and will decrease the amount of time spent on decision making. 

3. Subject selection - exclusion criteria: the authors excluded diffuse pulmonary disease such as ILD. What threshold they used for this exclusion? If patients had very subtle ILD (or ILA), were they excluded? And how could the authors know the existence of ILD? They reviewed all the CT exams? Please clarify.

All studies were initially selected based on the reports in the electronic health records. This provided us with a clinical diagnosis, including the presence of lung nodules or other concomitant lung diseases. If the diagnosis of a diffuse pulmonary disease, like ILD, was made in the clinical report, these studies were excluded. However, emphysema, likely from a smoking history, was We have clarified this in the materials and methods section, lines 121 to 123.

4. (There was no comment #4)

5. Subject selection- group categorization seems cumbersome for future readers and the authors described that Group 1 consisted of 178 CT scans reported as being free from pulmonary nodules. But the result (Table 1) is different. It was read that there was no pulmonary nodule, but were these cases actually had pulmonary nodules?

We thank the reviewer for this excellent question and observation. We selected CT studies based on the clinical report, but (almost as expected), some lung nodules had been missed and therefore were only reported during this study. As these were historical cases, we did not evaluate whether this could have had an impact on the patient outcome, given that malignancy should have become clear before the start of this selection process. We have clarified this in the materials and methods section, line 127 and 128 and in the results, lines 244 and 245. 

6. Only one CT machine was used in this study. If so, it seems very unified protocols and well organized study protocol. If not, please state more clear about types of CT machines.

This was a single center study, using historical cases. We selected a period of stability, and therefore the work was done on a single vendor system. We appreciate that this may be a weakness, and have highlighted this in the discussion. We have addressed this in the discussion section, lines 331 to 338. 

7. Nodule definition - please consider to delete "“pulmonary nodule” was not firmly defined since the notion of nodule may not represent a single entity capable of verbal definition." It seems unnecessary.

We thank the reviewer for this suggestion and have removed this sentence.

8. Nodule definition - the authors defined "actionable nodules" as a largest axial diameter between ≥5mm (or a volume of ≥80mm3) and ≤30mm. I wonder whether they have some references for this definition or they arbitrarily decided it.

“Actionable nodules” means nodules that require follow-up based on risk of malignancy. We have applied the British Thoracic Society guidelines on nodule management (reference: Callister et al., 2015), and have added this to the end of the sentence.

9. CAD software - For readers who are not friendly with this commercially available CAD system, please add detailed information about the CAD. For example, 1) why the threshold was decided as 0.1? what does it mean? 2) which CT examinations were used for its development in terms of slice thickness or use of contrast media?

Thank you for suggesting to add this information. Based on this comment and others comments from reviewer #1 and #2 we have adapted the chapter about the CAD software. 

In brief: the operating point was configured to yield a high sensitivity in order to detect as many actionable nodules as possible. Configuration favoring sensitivity automatically also yields a high number of false positives. 

The device was developed using a lung cancer screening cohort using appropriate screening scanning protocols which include slice thickness <=3mm and without contrast media. A part for conducting the study described in the manuscript was to investigate how the AI models (which were trained on screening scans) would relate to scans obtained in routine clinical practice.

10. Image annotation - "three different groups: micro-nodules, masses, benign nodules and non-nodules." Not three different groups. Please revise it.

Thank you for this remark, we will change the text to: “four different groups”. 

11. Reference standard and Data analysis -The description is rather complex and difficult to understand. I hope it is organized so that it is easy to understand.

We thank both reviewers for this comment. We have tried to clarify the process in the materials and methods section. The nodules were read independently, and blinded for clinical information, by two experienced chest radiologists, with a third senior chest radiologist available if no consensus was reached.

12. Reference standard and Data analysis -The most curious thing is that the authors created a reference standard as a result of the panels consisting of readers 1, 2, and 3. Nevertheless, it seems that the performance evaluation of readers was done as this reference standard.

The readers have read each case unaided and aided by the CAD (concurrent reader). As there were independent reads in place, we were able to construct a consensus report for all nodules, involved a third reader where necessary, which served as the reference standard.

13. Results: The content of the result may seem appropriate, but revision should be made according to the above items being modified.

We thank the reviewer for his/her helpful comments, and hope that the changes made according to the suggestion have clarified questions and improved the manuscript accordingly.

Response to Reviewer #2

We thank reviewer #2 for understanding the message of our manuscript by providing a correct and concise summary of the research and its objectives. 

Major comments of reviewer #2: 

1. The CAD system that is under investigation here is used for setting the reference standard. In addition, the readers for which the aided and unaided performance is reported, are used to make the reference standard. This affects the results and potentially positively biases the CAD results. The authors have also mentioned this in the Discussion so it is recognized as a limitation already. I think the conclusions of this study should therefore be less strong. It would be best if two other readers would also split all cases and read all of them without CAD support. Then, a truly independent read would be available. Ideally, this reader would read with and without CAD, but then, all cases need to be read twice, so is more effort.

We thank the reviewer for this comment. We wish to make it clear that the reference standard was created by a consensus report of up to three (where required) experienced chest radiologists. The software was indeed part of creating the reference” standard, but we opted to test the software by allowing readers to see 50% of cases with added software information to assess its impact. 

2. Related to the first point: A proper experiment setup for an observer study to compare aided vs unaided reads of data is multi-reader multi-case (MRMC) analysis, which also allows for better statistical comparisons than the t-tests that are performed now. If it is still possible to ask additional readers, that would make the study stronger. In addition, please consult with a statistician to see whether a form of MRMC analysis is possible on this data.

As stated under point 1, we only investigated the potential impact of the software tool once the readers had independently read the studies and reached consensus. We apologize this wasn’t made clear, and have amended the text.

3. The CAD is used at a setting with high sensitivity and relatively high false positive rates. If this is the setting that the radiologists used while reading the cases, then the paper should also report the standalone performance of the CAD system at this operating point. It is not clear to me what the performance is at this operating point. Several operating points are reported by the authors, but it is not clear to me which one correspond to the setting used by the readers (threshold of 0.1). Please add that as a dot on the FROC curve in Figure 1. Please also discuss what effect this setting may have on the study results in the Discussion part.

We thank the reviewer for this comment and have added this to the text, see lines 284 – 286. The standalone performance of the CAD, when set to 0.1, correlates to an average sensitivity of 95% and an average number of 7 false positives per study based on this dataset. 

4. Is the current CAD system approved/cleared for clinical use as a second reader or concurrent reader? This is not clear to me.

The current CAD system is CE certified as a second and concurrent reader.

5. Some important related literature in this area is not covered, for example: https://www.ajronline.org/doi/pdf/10.2214/AJR.17.18718

We thank the reviewer for this suggestion, and have incorporated this reference to the introduction and discussion.

6. A table which gives a breakdown of the CAD marks and the TP and FP categories would be very useful. The authors wrote "Finally, the readers classified all FP CAD prompts into three different groups: micro-nodules (largest axial diameter <3mm), masses (largest axial diameter >30mm), benign nodules (benign calcification pattern or clear benign perifissural appearance) and non-nodules (pleural plaque, scar tissue, atelectasis, fibrosis, fissure thickening, pleural fluid, pleural thickening, intrapulmonary vessels, consolidations, outside of lung tissue, or other (free format))." So, for every CAD mark, this rating is available so would be great to see a table with this information.

We agree that this is useful information to add to the manuscript, thank you for the suggestion. Instead of adding an extra table to the paper, we have added this information to the text in the M&M section on line 178 to 180. 

Detailed comments of reviewer #2:

7. Please be more clear about the selection of the CT cases. If I understand correctly, the original radiologist reports were manually checked to see whether nodules were reported or not, and that gave groups 1 and 2. But then the abstract should not state the exact nodule numbers yet, because I suppose this is the result after the annotation process, so should be in the Results section of the abstract.

Thank you for this suggestion, we have changed the abstract accordingly. We acknowledge that the total number of cases per group are part of the result section.

8. The statement on Data Availability: I understand that the scans cannot be shared, but it would be good if the result files with the readings can be shared. So, a csv file with for each cases the recorded nodules per reader, and whether they were in the end part of the reference standard, etc.

We have made a csv file with all raw data available and have uploaded this to the PLOS One site. 

9. How many CAD marks in total were there at the threshold of 0.1?

The standalone performance at a detection threshold of 0.1 are: 327 actionable nodules – 16 false negatives (FN) = 311 true positives (TP); 311 TP + 1927 false positives (FP) = 2238 CAD marks. 

10. How many CAD marks were there on micronodules or masses or marks only annotated by one reader and thus ignored in the FROC analysis? Please report.

Since the inclusion criteria were: all nodules ≥ 5 mm / ≥ 80 mm3 AND < 30 mm based on average segmentation. The following nodules were excluded based on the above criteria: nodules without majority consensus: 208; nodules < 5mm or < 80 mm3: 86; masses (≥ 30 mm): 6; benign nodules (calcified or perifissural nodules with actionable size: 59.

---

## [Decision Letter · Decision Letter 1]

19 Oct 2021

PONE-D-21-07234R1

Validation of a deep learning computer aided system for CT based lung nodule detection, classification, and growth rate estimation in a routine clinical population

PLOS ONE

Dear Dr. Nijwening,

Thank you for submitting your manuscript to PLOS ONE. After careful consideration, we feel that it has merit but does not fully meet PLOS ONE’s publication criteria as it currently stands. Therefore, we invite you to submit a revised version of the manuscript that addresses the points raised during the review process.

We look forward to receiving your revised manuscript.

Kind regards,

Chang Min Park, MD, Ph.D

Academic Editor

PLOS ONE

Reviewers' comments:

Reviewer's Responses to Questions

**Comments to the Author**

1. If the authors have adequately addressed your comments raised in a previous round of review and you feel that this manuscript is now acceptable for publication, you may indicate that here to bypass the “Comments to the Author” section, enter your conflict of interest statement in the “Confidential to Editor” section, and submit your "Accept" recommendation.

Reviewer #1: (No Response)

Reviewer #2: (No Response)

2. Is the manuscript technically sound, and do the data support the conclusions?

Reviewer #1: Partly

Reviewer #2: Partly

3. Has the statistical analysis been performed appropriately and rigorously? 

Reviewer #1: Yes

Reviewer #2: No

4. Have the authors made all data underlying the findings in their manuscript fully available?

Reviewer #1: No

Reviewer #2: Yes

5. Is the manuscript presented in an intelligible fashion and written in standard English?

Reviewer #1: Yes

Reviewer #2: Yes

6. Review Comments to the Author

Reviewer #1: Thank you for submitting the manuscript. I raised the following queries to improve the quality of this study

1. Page 16, lines 95-96: Please specify the diagnostic performance of radiologists (sub-specialist or specialist) for future readers to be interested in this manuscript and compare the performance with that of the CAD’s system.

2. Page 16, line 101-102: Please clarify how different malignancy probability pulmonary nodules have as they are solid or subsolid nodules presented in the CT examinations.

3. Page 18, line 117: Please give specific intervals the subjects participants in this study instead of “obtained at least 5 years prior to this study.”

4. The “subject selection” section is somewhat complicated. The authors should suggest a flowchart for selecting the study population as Figure.

5. As can be seen from the title, this study is a validation study in the routine clinical population. In fact, there are cases where there are more than 10 pulmonary nodules in routine clinical practice, and there is no other mention of this. Please explain this.

6. Please suggest all CT scanners included in this study.

7. There is no figure using the Veye Chest software in this study. Since most future readers are unfamiliar with this software, the authors will need a representative figure of how it works.

8. Why did the authors set the threshold of the software as 0.1? Of course, sensitivity is vital importance in the screening setting, the authors should perform various settings with various threshold values (for example, 0.1, 0.3, 0.5, 0.7) to adjust false-positive results.

9. “The detection results of CAD were made available at random in half the scans.” conflict to the following sentences. The CAD system was not all CT scans? Please revise this part appropriately.

10. Page 20, line 168-169: Please add the reference of the “Any nodules requiring follow-up according to lung cancer screening criteria were classified as “actionable nodules.”

11. Why the readers evaluated the CT features (solid or sub-solid) on a tablet? I believe that a dedicated workstation for reading CT scans is appropriate.

12. The concept of “center” of the nodule and adjudication of TP or FP with this concept is confusing. Please clarify and revise this part to make it easier to understand.

13. The authors mentioned that group 1 consisted of 178 CT scans being free from nodules. How was sensitivity calculated from group 1 without nodules?

14. Is it possible to statistically compare diagnostic performance between the CAD alone and radiologists with or without CAD?

Reviewer #2: I want to thank the authors for the additional effort that went into this manuscript, and for clarifying some of my comments.

I am however not satisfied with how the authors addressed my main points of criticism.

The reference standard is set by using the tested CAD system at a high sensitivity (and high FP rate) setting, and the same readers that set the reference standard are used for testing whether CAD helps them. This introduces a bias, even if there is a third experienced radiologist as an adjudicator. The authors have not addressed my suggestions for compensating for this. Another alternative solution that I did not mention before would be to take the full set of marks found by the two readers during the reading session (50% aided, 50% unaided) and have a new panel of 3 radiologists review this consolidated set of marks and set the reference standard. Note that this reference panel should then be blinded as to whether the mark that they are presented with was detected by a reader only, or by the reader following a CAD prompt.

Finally, the claim that the software helps radiologists is with the current data only partly supported when the CAD software is set at this high sensitivity threshold. Using the presented data, we cannot make conclusions as to whether the CAD software would still help radiologist when it is used at a different, more clinically acceptable operating point of 1 or 2FPs on average per scan.

7. PLOS authors have the option to publish the peer review history of their article (what does this mean?). If published, this will include your full peer review and any attached files.

Reviewer #1: No

Reviewer #2: No

---

## [Author Response · Author response to Decision Letter 1]

15 Dec 2021

Rebuttal letter, second response to reviewers

Reviewer #1: Thank you for submitting the manuscript. I raised the following queries to improve the quality of this study

1. Page 16, lines 95-96: Please specify the diagnostic performance of radiologists (sub-specialist or specialist) for future readers to be interested in this manuscript and compare the performance with that of the CAD’s system.

As you suggested, we have added the corresponding performance from the cited papers to the text. However, we cannot compare this performance to what we find in our study (with and without CAD support) because the introduction is not the place to add this. We do compare our results with that of other studies in the Discussion section. 

2. Page 16, line 101-102: Please clarify how different malignancy probability pulmonary nodules have as they are solid or subsolid nodules presented in the CT examinations.

We thank the reviewer for this observation and have added to the manuscript that sub-solid nodules are likely to be malignant (see Track Changes). In the current guidelines there is not yet a clear cut off or probability factor for malignancy. Sub-solid nodules are more often identified as malignant when analyzed (after a biopsy / resection, or via PET). The Fleischner recommendation in the cited paper states the following about sub-solid nodules: “Solitary part-solid GGNs, especially those in which the solid component is larger than 5 mm, should be considered malignant until proved otherwise provided either growth or no change is seen at a follow-up CT examination performed in 3 months”.

3. Page 18, line 117: Please give specific intervals the subjects participants in this study instead of “obtained at least 5 years prior to this study.”

Thank you for this remark. Based on this and your following comment, we have rewritten the whole “Subject Selection” part to be more clear about the procedure of collecting the scans for the study, see the rewritten part in Track Changes. In addition, we would like to add that CT studies between January 2008 and December 2009 were searched for potential inclusion into this study. This gave us a follow-up period of more than 7 years, to enable potential reference standard to be assessed.

4. The “subject selection” section is somewhat complicated. The authors should suggest a flowchart for selecting the study population as Figure.

This is a good suggestion. We realize that the “Subject Selection” paragraph could be confusing and we would like to suggest to rewrite this paragraph in a concise way instead of adding a flowchart as an extra figure to the text. We hope that the reviewer and editor agree to this suggestion. Please find the rewritten part in the uploaded “Revised Manuscript with Track Changes”. 

5. As can be seen from the title, this study is a validation study in the routine clinical population. In fact, there are cases where there are more than 10 pulmonary nodules in routine clinical practice, and there is no other mention of this. Please explain this.

This is a good question and we understand why the reviewer is asking this since, indeed, we see cases with more than 10 nodules in clinical practice. For this study, however, we choose for simplicity reasons to cap the maximum amount of nodules in a study to 10. Since in our population the vast majority of patients has less than 10 nodules. Adding studies of more than 10 nodules per patient would not add more value to the quality of the study. 

6. Please suggest all CT scanners included in this study.

These are the number of studies performed with a specific CT scanner: 

• Toshiba 333 Aquillion: 330

• Toshiba Aquillion-CX: 2

• Toshiba Aquillion ONE: 1

• GE Medical Systems LightSpeed 16: 2

• GE Medical Systems LightSpeed: 2

We have added this to the revised manuscript. 

7. There is no figure using the Veye Chest software in this study. Since most future readers are unfamiliar with this software, the authors will need a representative figure of how it works.

We would like to thank the reviewer for this excellent suggestion. We have added a screenshot of the actual product to the supplementary material. I do want to stress, however, that the intention of this manuscript is non-promotional and therefore we want to exclude branded material as much as possible from the main body of text and figures. 

8. Why did the authors set the threshold of the software as 0.1? Of course, sensitivity is vital importance in the screening setting, the authors should perform various settings with various threshold values (for example, 0.1, 0.3, 0.5, 0.7) to adjust false-positive results.

We would like to thank the reviewer for this question and expect that the following answer suffices. As discussed in the first paragraph in the Discussion, the threshold of 0.1 was based on a sensitivity of 95% and a false-positive rate of 7, which is a workable number of false-positives. We have done a variation of the thresholds, as demonstrated in the range of threshold of false positive lung nodules in the results section. In addition, the nodule software was not used as a screening tool, rather for evaluation of incidental pulmonary nodules in a routine chest CT population using standard-dose CT protocol. In this setting, it is commonplace to evaluate at a threshold of 80 mm3 or 5 mm detection level, and for this threshold, the software setting of 0.1 is the optimal setting. 

9. “The detection results of CAD were made available at random in half the scans.” conflict to the following sentences. The CAD system was not all CT scans? Please revise this part appropriately.

Thank you for this remark. We realize the set-up of our study could be more clarified and have adjusted the manuscript accordingly. Briefly, the CT-studies were divided in two parts, A and B. Reader 1 analyzed the studies in part A with CAD and part B without CAD. For Reader 2 this was the other way around: studies in part A were analyzed without CAD and part B with CAD. Discrepancies were compared. Reader 3 subsequently adjudicated all discrepancies without the results of CAD.

10. Page 20, line 168-169: Please add the reference of the “Any nodules requiring follow-up according to lung cancer screening criteria were classified as “actionable nodules.”

Thank you for this suggestion, we have added reference 10 behind this sentence because we are referring to the same reference as is cited in the Nodule Definition paragraph in the Materials and Methods section. 

11. Why the readers evaluated the CT features (solid or sub-solid) on a tablet? I believe that a dedicated workstation for reading CT scans is appropriate.

Good question from the reviewer, we would like to give the following explanation: “We used a high-definition iPAD Pro tablet in order to facilitate this study, which was performed by three different radiologists. Although possibly slightly suboptimal compared to a dedicated workstation, this pragmatic approach was chosen in order to allow this study to take place outside the working environment. We don’t believe that this significantly altered the outcome of the study, as we were able to compare nodules detected by the study radiologists with clinical reports (and indeed found more nodules than initially reported).”

12. The concept of “center” of the nodule and adjudication of TP or FP with this concept is confusing. Please clarify and revise this part to make it easier to understand.

We thank the reviewer for this question and believe that revision of the manuscript is not necessary because this is a standard method. The CAD software should position and prompt within the center of any nodule, and we wished to find out of this prompt aligned with the mark-up of the two radiologists. We then applied the Dice coefficient method to detect discrepancies in the segmentation between the software and the various observers. This method is described in the methods section, and is a common way of demonstrating the level of overlap between the different segmentations in these types of studies.

13. The authors mentioned that group 1 consisted of 178 CT scans being free from nodules. How was sensitivity calculated from group 1 without nodules?

We can be very brief answering this question: sensitivity is calculated on lesion level, not on scan level. We hope the reviewer will be satisfied with this answer. 

14. Is it possible to statistically compare diagnostic performance between the CAD alone and radiologists with or without CAD?

Yes, this was the set-up of our study as described in the re-written part of the “Image Annotation” chapter of the Materials and Methods section. We have rewritten this part as per suggestion of the reviewer (comment #9). We hope we made this section more clear and thank the reviewer for notifying the opaqueness of this part. 

Reviewer #2: I want to thank the authors for the additional effort that went into this manuscript, and for clarifying some of my comments.

I am however not satisfied with how the authors addressed my main points of criticism.

The reference standard is set by using the tested CAD system at a high sensitivity (and high FP rate) setting, and the same readers that set the reference standard are used for testing whether CAD helps them. This introduces a bias, even if there is a third experienced radiologist as an adjudicator. The authors have not addressed my suggestions for compensating for this. Another alternative solution that I did not mention before would be to take the full set of marks found by the two readers during the reading session (50% aided, 50% unaided) and have a new panel of 3 radiologists review this consolidated set of marks and set the reference standard. Note that this reference panel should then be blinded as to whether the mark that they are presented with was detected by a reader only, or by the reader following a CAD prompt.

Answer to Reviewer #2: We would like to thank Reviewer #2 for her/his critical appraisal of the manuscript. In this and the previous review-round, Reviewer #2 suggests to perform additional studies with more readers to confirm the results described in this manuscript. However, we choose not to invest in further analyses of a study that already has been completed, but to invest in additional research questions instead. There are more studies with the same software ongoing and published, please search for “Aidence” in PubMed. 

In this manuscript, we describe the validation of software, that was initially developed with lung cancer screening data, in “everyday clinical use”. In this routine clinical practice we aim for high detection rates of nodules because of their potential malignancy. Therefore, we apply a CAD system with high sensitivity. Radiologists were allowed to accept or refute marks indicating potential nodules. This is a commonly applied methodology, as it allows the detection of nodules otherwise missed, while the radiologist can discount “over call”. This is how a system works in clinical practice when aiming for high detection rates: high sensitivity at a cost of decreased specificity. Examples include mammography screening and also screening tests such as cervical smear testing and occult blood testing for colon cancer.

We do acknowledge the concern of reviewer #2 and have addressed this as a limitation of our study in the discussion. At the time of the first clinical validation of our novel software, time and budget were limited. However, we believe that the study described in this manuscript is fair and significant, despite the constraints we had to face when conducting it. 

Reviewer #2: Finally, the claim that the software helps radiologists is with the current data only partly supported when the CAD software is set at this high sensitivity threshold. Using the presented data, we cannot make conclusions as to whether the CAD software would still help radiologist when it is used at a different, more clinically acceptable operating point of 1 or 2FPs on average per scan.

Answer to Reviewer #2: We thank the reviewer again and we respectfully disagree. What the reviewer claims as clinically acceptable is actually very close to what we were seeing in the aided reads: an average FP rate of 0.11 (radiologists without CAD) and 0.16 (radiologists with CAD) per CT scan. This merely reinforces what we stated above: we require high sensitivity settings in order not to miss actionable nodules. We have addressed the limitations of the study in our discussion and we believe that we validated the point of our retrospective study: showing that this software could help radiologists in their daily clinical practice to not miss pulmonary nodules in patients.

---

## [Decision Letter · Decision Letter 2]

4 Feb 2022

PONE-D-21-07234R2

Validation of a deep learning computer aided system for CT based lung nodule detection, classification, and growth rate estimation in a routine clinical population

PLOS ONE

Dear Dr. Nijwening,

Thank you for submitting your manuscript to PLOS ONE. After careful consideration, we feel that it has merit but does not fully meet PLOS ONE’s publication criteria as it currently stands. Therefore, we invite you to submit a revised version of the manuscript that addresses the points raised during the review process.

ACADEMIC EDITOR: Please insert comments here and delete this placeholder text when finished. Be sure to:

First of all, thank you very much for your time and effort for the revision.

Your paper showed much improvement through the revision, but I felt it still needs additional work for the final acceptance. You might want to check the reviews from our reviewers.

We look forward to receiving your revised manuscript.

Kind regards,

Chang Min Park, MD, Ph.D

Academic Editor

PLOS ONE

Journal Requirements:

Reviewers' comments:

Reviewer's Responses to Questions

**Comments to the Author**

1. If the authors have adequately addressed your comments raised in a previous round of review and you feel that this manuscript is now acceptable for publication, you may indicate that here to bypass the “Comments to the Author” section, enter your conflict of interest statement in the “Confidential to Editor” section, and submit your "Accept" recommendation.

Reviewer #1: All comments have been addressed

Reviewer #2: (No Response)

2. Is the manuscript technically sound, and do the data support the conclusions?

Reviewer #1: Partly

Reviewer #2: Partly

3. Has the statistical analysis been performed appropriately and rigorously? 

Reviewer #1: Yes

Reviewer #2: No

4. Have the authors made all data underlying the findings in their manuscript fully available?

Reviewer #1: Yes

Reviewer #2: Yes

5. Is the manuscript presented in an intelligible fashion and written in standard English?

Reviewer #1: Yes

Reviewer #2: Yes

6. Review Comments to the Author

Reviewer #1: First of all, thank you for your submitting your manuscript with appropriate revision. I think your revision was very appropriate and will make future readers’ concerns alleviated.

Reviewer #2: I want to thank the authors for their responses.

The rebuttal letter reads that the authors respectfully disagree with my statement "we cannot make conclusions as to whether the CAD software would still help radiologist when it is used at a different, more clinically acceptable operating point of 1 or 2FPs on average per scan." Based on the response in the letter, it is my understanding that the authors disagree that 1 or 2 FPS on average per scan is a more clinically acceptable operating point? This surprises me because the authors write in the discussion: "A more acceptable average FP rate would be between 1 and 2 with corresponding sensitivity range (82.3% - 89.0%), outperforming thoracic radiologists with and without using CAD."

So, I think the authors and me actually agree on what is a clinically acceptable setting, so I do not fully understand the response. My main point is that this study shows the CAD system set to operate at 7 FPs per scan helps increasing sensitivity of radiologists, but has no direct evidence what the added performance would be when the CAD is set to operate at 1 or 2 FPs per scan. Do the authors disagree with that?

The authors have decided to not do additional analysis and I respect that. I understand that priorities shift over time and that time and resources are limited. The concerns that I described are to a certain extent covered in the Discussion section.

If this paper is to be published in its current form, I think it is necessary to at least perform the following changes:

- As discussed in the first paragraph in the Discussion, the threshold of 0.1 was used in this study for the CAD, corresponding to a sensitivity of 95% and a false-positive rate of 7 on this dataset.

Since the CAD is used by the readers at this operating point, I think the result section of the abstract should report this performance, and not the performance at another operating point. Especially because the next sentences are about radiologist performance with or without CAD. That is confusing and misleading, in my opinion.

So, I think the first sentence of the Result section of the Abstract should read:

"After analysis of 470 pulmonary nodules, the sensitivity of CAD as a stand-alone test for detecting nodules was 95% with an average FP rate of 7 per CT scan at the operating point used in this study."

- The discussion section reads "These findings compare favorably with several other software tools (21,23)." I think this statement is not fair and not proven by the current results and with the current reference standard. Please remove this part.

7. PLOS authors have the option to publish the peer review history of their article (what does this mean?). If published, this will include your full peer review and any attached files.

Reviewer #1: No

Reviewer #2: No

---

## [Author Response · Author response to Decision Letter 2]

9 Feb 2022

9 February 2022

Reviewer #1: First of all, thank you for your submitting your manuscript with appropriate revision. I think your revision was very appropriate and will make future readers’ concerns alleviated.

Corresponding author: thank you for your contributions and critical appraisal of our manuscript, reviewer #1.

Reviewer #2: I want to thank the authors for their responses.

The rebuttal letter reads that the authors respectfully disagree with my statement "we cannot make conclusions as to whether the CAD software would still help radiologist when it is used at a different, more clinically acceptable operating point of 1 or 2FPs on average per scan." Based on the response in the letter, it is my understanding that the authors disagree that 1 or 2 FPS on average per scan is a more clinically acceptable operating point? This surprises me because the authors write in the discussion: "A more acceptable average FP rate would be between 1 and 2 with corresponding sensitivity range (82.3% - 89.0%), outperforming thoracic radiologists with and without using CAD."

So, I think the authors and me actually agree on what is a clinically acceptable setting, so I do not fully understand the response. My main point is that this study shows the CAD system set to operate at 7 FPs per scan helps increasing sensitivity of radiologists, but has no direct evidence what the added performance would be when the CAD is set to operate at 1 or 2 FPs per scan. Do the authors disagree with that?

Corresponding author: first of all, we (the corresponding author and the radiologists involved in the study) would like to thank reviewer #2 for the discussion and critical comments regarding our manuscript. In the end, this is what creates progress in science. Regarding your point above, we do not disagree. The primary purpose of our study was to validate the detection, segmentation, classification, and growth assessment of our software device in clinical practice. Not the detection, nor its impact on reader performance. The operating point was therefore set to an extreme level to make sure that the ground truth was as complete as possible. We agree with reviewer #2 that we would never use an OP of 7 FP per scan in clinical practice. We therefore will adopt the manuscript as suggested. 

Reviewer #2: The authors have decided to not do additional analysis and I respect that. I understand that priorities shift over time and that time and resources are limited. The concerns that I described are to a certain extent covered in the Discussion section.

Corresponding author: we would like to thank reviewer #2 for her/his understanding of shifting our priorities. We have invested our resources in an FDA study for 510(k) approval, of which we will publish the results as well. 

Reviewer #2: If this paper is to be published in its current form, I think it is necessary to at least perform the following changes:

- As discussed in the first paragraph in the Discussion, the threshold of 0.1 was used in this study for the CAD, corresponding to a sensitivity of 95% and a false-positive rate of 7 on this dataset.

Since the CAD is used by the readers at this operating point, I think the result section of the abstract should report this performance, and not the performance at another operating point. Especially because the next sentences are about radiologist performance with or without CAD. That is confusing and misleading, in my opinion.

So, I think the first sentence of the Result section of the Abstract should read:

"After analysis of 470 pulmonary nodules, the sensitivity of CAD as a stand-alone test for detecting nodules was 95% with an average FP rate of 7 per CT scan at the operating point used in this study."

Corresponding author: in concordance with the points you raised above and in previous review rounds, we agree with you. Therefore, we have decided to completely remove the sensitivity and FP numbers from the abstract (please see the revised manuscript with track changes). After all, the purpose of the study was to validate an algorithm and therefore the radiologist performance with or without CAD is the main point of the study. 

Reviewer #2: The discussion section reads "These findings compare favorably with several other software tools (21,23)." I think this statement is not fair and not proven by the current results and with the current reference standard. Please remove this part.

Corresponding author: yes, we agree with this suggestion and have removed the sentence from the manuscript.

---

## [Editor Report · Decision Letter 3]

14 Mar 2022

PONE-D-21-07234R3Validation of a deep learning computer aided system for CT based lung nodule detection, classification, and growth rate estimation in a routine clinical populationPLOS ONE

Dear Dr. Nijwening,

Thank you for submitting your manuscript to PLOS ONE. After careful consideration, we feel that it has merit but does not fully meet PLOS ONE’s publication criteria as it currently stands. Therefore, we invite you to submit a revised version of the manuscript that addresses the points raised during the review process.

We look forward to receiving your revised manuscript.

Kind regards,

Chang Min Park, MD, Ph.D

Academic Editor

PLOS ONE

Journal Requirements:

Additional Editor Comments:

Dear Authors:

Thank your for your rebuttal and revision. I can understand your points. However, there still remains several minor parts that needs revisions for final acceptance.

Fortunately, I think you can respond to them without efforts.  

1. line 34-35: "the readings for radiologists without CAD and radiologist with CAD, were 71.9% ... and 80.3%.."  The meaning of each number is not clear by this sentence, and thus should be properly revised.

2. line 45-47: "Results suggest this software could assist chest radiologists in pulmonary nodule detection and management within their routine clinical practice." I think this statement cannot be supported by the results, so I recommend to remove this part or the authors might want to revise as "The deep Learning software has the potential to assist radiologists in the tasks of pulmonary nodule detection and management on routine chest CT, which was written in the Discussion section by the authors.

3. line 82-84: "Lung cancer remains the third most prevalent cancer worldwide, is both rising in incidence (1), and maintains high mortality rates with around 1.7 million global deaths annually."

 Reference #1 is too much out-of-date. The authors need to update the reference.

4. line 290-292: "the results of our study are of similar sensitivity and accuracy to that initial cohort, (87% at 1 FP/scan) confirming broader use is feasible." This part needs a reference.

5. line 337-340: "The Deep Learning model for nodule detection was trained on data from a lung cancer screening cohort but this study shows that it is effective in a general, “real life” clinical setting where it improves the sensitivity of detection of actionable nodules by thoracic radiologists."  this statement went too far and actually was not supported by the results. So the authors might want to remove this part.
---

## [Author Response · Author response to Decision Letter 3]

25 Mar 2022

Rebuttal Letter, 4th response to reviewers

25 March 2022

Reviewer: 1. line 34-35: "the readings for radiologists without CAD and radiologist with CAD, were 71.9% ... and 80.3%.."  The meaning of each number is not clear by this sentence, and thus should be properly revised.

Corresponding author: thank you for highlighting this error, the numbers indicate sensitivity and we have added this to the text. 

Reviewer: 2. line 45-47: "Results suggest this software could assist chest radiologists in pulmonary nodule detection and management within their routine clinical practice." I think this statement cannot be supported by the results, so I recommend to remove this part or the authors might want to revise as "The deep Learning software has the potential to assist radiologists in the tasks of pulmonary nodule detection and management on routine chest CT, which was written in the Discussion section by the authors.

Corresponding author: thank you for your suggestion, we have adapted the text accordingly, see “track changes”. 

Reviewer: 3. line 82-84: "Lung cancer remains the third most prevalent cancer worldwide, is both rising in incidence (1), and maintains high mortality rates with around 1.7 million global deaths annually."

 Reference #1 is too much out-of-date. The authors need to update the reference.

Corresponding author: the reviewer is right, new data indicates that lung cancer mortality rates are even 1.8 million global deaths. We have edited the text and the reference accordingly. 

Reviewer: 4. line 290-292: "the results of our study are of similar sensitivity and accuracy to that initial cohort, (87% at 1 FP/scan) confirming broader use is feasible." This part needs a reference.

Corresponding author: we have added the reference. 

Reviewer: 5. line 337-340: "The Deep Learning model for nodule detection was trained on data from a lung cancer screening cohort but this study shows that it is effective in a general, “real life” clinical setting where it improves the sensitivity of detection of actionable nodules by thoracic radiologists."  this statement went too far and actually was not supported by the results. So the authors might want to remove this part.

Corresponding author: we have altered the original sentences, see track changes, and weakened our initial conclusion. We expect that these new sentences will be in line with what the reviewer suggests.

---

## [Editor Report · Decision Letter 4]

29 Mar 2022

Validation of a deep learning computer aided system for CT based lung nodule detection, classification, and growth rate estimation in a routine clinical population

PONE-D-21-07234R4

Dear Dr. Jeroen Nijwening:

We’re pleased to inform you that your manuscript has been judged scientifically suitable for publication and will be formally accepted for publication once it meets all outstanding technical requirements.

Kind regards,

Chang Min Park, MD, Ph.D

Academic Editor

PLOS ONE

---

## [Editor Report · Acceptance letter]

26 Apr 2022

PONE-D-21-07234R4 

Validation of a deep learning computer aided system for CT based lung nodule detection, classification, and growth rate estimation in a routine clinical population 

Dear Dr. Nijwening:

I'm pleased to inform you that your manuscript has been deemed suitable for publication in PLOS ONE. Congratulations! Your manuscript is now with our production department. 

Kind regards, 

on behalf of

Professor Chang Min Park 

Academic Editor

PLOS ONE